# *Duguetia lanceolata* A. St.-Hil. (Annonaceae) Essential Oil: Toxicity against *Spodoptera frugiperda* (J. E. Smith) (Lepidoptera: Noctuidae) and Selectivity for the Parasitoid *Trichogramma pretiosum* Riley (Hymenoptera: Trichogrammatidae)

**Mayara Ketllyn de Paula Rosetti** [1]**, Dejane Santos Alves** [1,*]**, Isabela Caroline Luft** [1]**, Katiane Pompermayer** [1]**, Andressa Soares Scolari** [1]**, Gabriela Trindade de Souza e Silva** [2]**, Murilo Silva de Oliveira** [1]**, Javier Andrés García Vanegas** [3]**, Horácio Bambo Pacule** [3]**, Geraldo Humberto Silva** [4]**, Denilson Ferreira de Oliveira** [3] **and Geraldo Andrade Carvalho** [5,*]

[1] Campus Santa Helena (UTFPR-SH), Universidade Tecnológica Federal do Paraná, Santa Helena 85892-000, Paraná, Brazil

[2] Departamento de Ciências Farmacêuticas, Universidade Estadual de Campinas, Campinas 13083-887, São Paulo, Brazil

[3] Departamento de Química, Universidade Federal de Lavras (UFLA), Lavras 37200-000, Minas Gerais, Brazil

[4] Instituto de Ciências Exatas, Universidade Federal de Viçosa, Rio Parnaíba 38810-000, Minas Gerais, Brazil

[5] Departamento de Entomologia, Universidade Federal de Lavras (UFLA), Lavras 37200-000, Minas Gerais, Brazil

**\*** Correspondence: dejanealves@utfpr.edu.br (D.S.A.); gacarval@ufla.br (G.A.C.); Tel.: +55-35-9-9125-4032 (D.S.A.); +55-35-9-8817-9756 (G.A.C.)

**Abstract:** The fall armyworm (FAW) *Spodoptera frugiperda* is a polyphagous pest that is difficult to control due to populations resistant to various active ingredients. Thus, the objective of this study was to evaluate the toxicity of essential oils (EOs) from the organs of *Annona neolaurifolia*, *Duguetia lanceolata*, and *Xylopia brasiliensis*, against the FAW and its natural enemy, *Trichogramma pretiosum*. The most active EOs were those from the leaves and stem bark of *D. lanceolata*, which presented $LD_{90}$ to *S. frugiperda* equal to 70.76 and 127.14 μg of EO/larvae, respectively. The major compounds in the EO of *D. lanceolata* (leaves) were β-caryophyllene and caryophyllene oxide. Although individually inactive against the FAW, when combined, those compounds reduced the insect's probability of survival. However, the mortality was lower than that caused by EO. This result suggests that other components of EO contribute to the activity against FAW. Furthermore, the EO of the leaves from *D. lanceolata* presented low toxicity to the egg-larva stage of *T. pretiosum*, but was toxic to other phases. Thus, EO from *D. lanceolata* is potentially useful for developing new products to control *S. frugiperda*.

**Keywords:** botanical insecticide; physiological selectivity; (*E*)-caryophyllene; caryophyllene epoxide

## 1. Introduction

The fall armyworm (FAW) *Spodoptera frugiperda* (J. E. Smith, 1797) (Lepidoptera: Noctuidae) is a highly polyphagous insect with great dispersion capacity [1]. Although it is originally from the Americas, it has dispersed to Africa [2], Asia [3], and Oceania [4]. Recently, the FAW was detected in the Mediterranean region, specifically in Spain [4]. Notably, the entry of FAW into underdeveloped countries, such as those on the African continent, led to speculation about a global crisis in food production [5]. The feeding behavior of FAW has changed over time; although able to feed on more than 350 plant species, it prefers maize [1]. Initially, this insect used to feed preferentially on the corn cartridge, but now it can attack all stages of the crop, from the seedling to the cob, so that it can cause up to 100% crop loss when control measures are not used [6,7].

Synthetic chemical insecticides and genetically modified plants are fundamental tools and the most used methods to keep the FAW population below an economic damage level.

However, due to the intensive use of these tools, increasing reports on the selection of insect populations resistant to these insecticides that feed on genetically modified plants have emerged [8–11]. Therefore, there is an urgent need to search for new molecules to be used in FAW control. One of the existing possibilities to satisfy this need concerns botanical insecticides, which have had a resurgence in recent years due to the easing of regulatory approvals in developed countries [12], and restrictions imposed on some synthetic chemical insecticides [13]. Among the groups of botanical insecticides, the commercialization of EOs has been highlighted, as these are often commercialized in binary mixtures or sometimes tertiary. Large companies with sizable shares of the chemical insecticide market have invested in developing and trading botanical insecticides. For example, the insecticide Requiem® (Bayer Crop Science) is mainly composed of EO constituents from *Dysphania ambrosioides* (L.) Mosyakin and Clemants (Amaranthaceae). The major constituents: α-terpinene, p-cymene, and (R)-limonene, are obtained by the synthesis in the laboratory to make the product economically viable [14].

Among the botanical families that have the potential for producing secondary metabolites with insecticidal activity, the Annonaceae family can be highlighted [15]. In this family, the species: *Annona neolaurifolia* H. Rainer (Annonaceae), *Duguetia lanceolata* A. St.-Hil. (Annonaceae), and *Xylopia brasiliensis* R.E.Fr (Annonaceae) have the potential for developing new insecticides but are yet to be explored. The toxicity to the FAW by other species of the genus *Annona* has already been reported. However, studies were conducted with plant extracts, and the insecticidal activity was attributed to the presence of acetogenins [16–19]. The fraction soluble in dichloromethane from the methanolic extract of the species *D. lanceolata* has already been reported to have insecticidal activity in an ingestion test with FAWs [20,21]. However, studies employing the EOs of this plant against FAWs were not observed. Regarding the genus *Xylopia*, it is worth mentioning that the EOs of *Xylopia aethiopica* (Dun) A. Rich. (Annonaceae) was toxic, in a topical application test, to *Spodoptera littoralis* (Boisduval) (Lepidoptera: Noctuidae) [22].

In addition to obtaining efficient products to control the insect pest, it is crucial to ensure that such products have the least possible effect on non-target organisms, especially when such organisms may be natural enemies of the insect pest. This scenario is the case for the egg parasitoid *Trichogramma pretiosum* Riley (Hymenoptera: Trichogrammatidae), a natural enemy of the FAW. Egg parasitoids of the genus *Trichogramma* are among the most used biological control agents in the world due to the ease of rearing in alternative hosts at a competitive cost [23]. In China, massive investments in production technologies made this country a world leader in biological control programs employing *Trichogramma* spp. There are reports that just one production line in China can supply up to 200 billion parasitoids annually [24]. Moreover, in Brazil, it is estimated that 2,252,900 ha were treated with *Trichogramma* spp., highlighting the importance of this parasitoid [25].

Given the above, we hypothesized that the EOs from *A. neolaurifolia*, *D. lanceolata*, and *X. brasiliensis* are toxic to the FAW and not toxic to the parasitoid *T. pretiosum*. Therefore the objectives of this work were: (i) to assess acute and chronic toxicity (in free-choice and no-choice tests) of EOs from *A. neolaurifolia* (leaves, branches, and stem barks), *D. lanceolata* (leaves, branches, and stem barks), and *X. brasiliensis* (stem barks and branches) against *S. frugiperda*; (ii) to carry out the chemical characterization of the EOs; (iii) to determine the toxicity of the major compounds of the most active EO to the FAW, and (iv) to study the selectivity of the EOs, more toxic for the FAW, for the parasitoid *T. pretiosum*.

## 2. Materials and Methods

### 2.1. Plant Material and EOs

Botanical materials from *A. neolaurifolia* (leaves, branches, and stem barks), *D. lanceolata* (leaves, branches, and stem barks), and *X. brasiliensis* (stem barks and branches) were collected in February 2019 in the Alto do Rio Grande Region, Lavras, Minas Gerais, Brazil. The identification of the botanical material was conducted by botanists from the Esal Herbarium, Federal University of Lavras. Exsiccata were deposited in the Esal Herbarium. The fresh

material was submitted to hydrodistillation in an adapted Clevenger apparatus. The EOs were separated from the water by decantation, and anhydrous sodium sulfate was used to remove traces of water from the oils. As the EOs of *A. neolaurifolia* (leaves), *D. lanceolata* (leaves and stem barks), and *X. brasiliensis* (branches and stem barks) were obtained in higher yields, they underwent the subsequent stages of the present work (Table 1).

**Table 1.** Species of plants utilized to obtain essential oils.

| Species/Synonyms | Mass of Fresh Material (kg) | Collected Part | Oil Mass (g)/yield (%) | Number of Exsiccate | Geographic Coordinates |
|---|---|---|---|---|---|
| *Annona neolaurifolia* H. Rainer Sin. *Rollinia laurifolia* Schlt [26] | 2.2 1.1 1.2 | Leaves Stem barks Branches | 3.69 (0.19%) 0 0 | 27,638 | S 21°13.648′; W 044°57.405′ |
| *Duguetia lanceolata* A. St.-Hil. Sin. *Aberemoa lanceolata* var. *parvifolia* R.E.Fr. [27] | 1.9 4.3 1.1 | Leaves Stem barks Branches | 3.69 (0.19%) 2.54 (0.058%) 0 | 27,629 | S 21°13.567′; W 044°57.575′ |
| *Xylopia brasiliensis* R.E.Fr Sin. *Xylopia gracilis* (R.E.Fr.) R.E.Fr [26] | 2.9 1.4 | Stem barks Branches | 0.13 (0.009%) 0.82 (0.028%) | 27,636 | S 21°13.732′; W 044°58.064′ |

### *2.2. Chemical Characterization of the EOs*

The analyzes were conducted in a gas chromatograph coupled to a mass spectrometer (model QP2010, Shimadzu, Kyoto, Japan), equipped with an RTX®-5MS capillary column (30 m × 0.25 mm ID × 0.25 µm film thickness: Restek). The carrier gas used was helium at 1.0 mL min$^{-1}$. According to [28], the following conditions were used: (1) split/splitless injector temperature: 220 °C; (2) split ratio: 1:20; (3) initial column temperature: 60 °C; (4) rate of column temperature rise: 2 °C min$^{-1}$ to 200 °C and then 5 °C min$^{-1}$; (5) final column temperature: 250 °C; (6) interface temperature between gas chromatograph and mass spectrometer: 220 °C; (7) electron impact at 70 eV; (8) mass/charge (*m/z*) analysis range: 45–400 and (9) mass spectrum acquisition time: 0.5 s.

A solution of standard homologous linear alkanes containing nine to 20 carbon atoms was used as an external standard. All mass spectra were compared to the NIST 05 library 2005, and all peaks in the chromatogram with a similarity index below 90% were not identified. For each of the remaining peaks, the arithmetic index (AI) was calculated according to the following equation:

$$AI = 100Pz + 100 \left[ \frac{RT - RTPz}{RTPz + 1 - RTPz} \right] \tag{1}$$

where Pz is the number of carbon atoms of the linear alkane with a retention time immediately lower than that of the substance to be identified in the chromatogram; RT is the retention time (min) of the substance to be identified in the chromatogram; RTPz is the retention time (min) of linear alkane with the number of carbon atoms equal to Pz; and RTPz + 1 is the retention time (min) of linear alkane with the number of carbon atoms equal to Pz + 1. Compounds with calculated AI values corresponding to an error ≥3% in relation to the AI described by [28] were considered unidentified.

### *2.3. Bioassays with the FAW*
#### 2.3.1. FAW Maintenance Creation

Approximately 200 unsexed adult FAWs were kept in cages (19 cm wide × 20 cm high) and fed with aqueous honey solution 10% (*v/v*) ad libitum. The caterpillars were fed on an artificial diet: beans (166.7 g); wheat germ (79.20 g); brewer's yeast (50.70 g); sorbic acid PA (1.65 g); ascorbic acid PA (5.10 g); methylparaben (3.15 g); agar (27 g); formaldehyde (4.15 mL); and 4.15 mL of an inhibitor solution, composed by propionic acid (18 mL), phosphoric acid (42 mL) and water (540 mL) [29]. Insect maintenance breeding and

bioassays were conducted in a 25 ± 2 °C air-conditioned room, with a relative humidity of 70 ± 10% and a 12L:12D photoperiod.

### 2.3.2. Acute Toxicity of EOs against the FAW in a Topical Application Test

FAW caterpillars, previously maintained on an artificial diet (72-hour-old and 0.4 mm in length), were used. The EOs of *A. neolaurifolia* (leaves), *D. lanceolata* (leaves and stem barks), and *X. brasiliensis* (branches and stem barks) (10 mg) were dissolved in acetone (100 μL). Aliquots (1 μL) of the resulting solutions were applied to the back of the caterpillars, corresponding to a dose of 100 μg of EO/larvae. After application, the caterpillars were transferred to glass tubes (2.5 cm × 8.0 cm high) containing a piece of artificial diet (1.0 cm × 1.5 cm high). The tubes were closed with hydrophilic cotton plugs.

The assay was conducted in a completely randomized design with 50 replicates per treatment. Each replicate consisted of a tube containing a caterpillar. The negative control was acetone. The bioassay was repeated twice on different days. Insect survival was evaluated 24, 48, 72, 96, 120, 144, and 168 h after EO application. An insect that did not respond to the touch of a fine-bristled brush was considered dead.

### 2.3.3. Dose-Response and Time-Response Curves

The EOs from the leaves and stem bark of *D. lanceolata* was selected for this assay because they exhibited the highest activity against *S. frugiperda* in an acute toxicity test after topical application (Section 2.3.2) and high extraction yields (Table 1). The doses were determined through previous assays to obtain ranges that cause mortality from 20–80% of the insects [30]. The EOs from the stem bark of *D. lanceolata* were used in doses of 5, 9, 15, 27, and 50 μg of EO/larvae. On the other hand, the EOs from leaves of *D. lanceolata* were applied in doses of 1, 3, 10, 32, and 100 μg of EO/larvae. The commercial insecticide cypermethrin (Cypermethrin Pestanal®, analytical standard-Sigma-Aldrich®, St. Louis, MO, USA) was used as a positive control in the doses of 0.0001, 0.0017, 0.032, 0.5623, and 10 μg of active ingredient (a.i.)/larvae. All treatments were applied topically to the insects, as described in Section 2.3.2.

The design was completely randomized, with 50 repetitions per treatment, each formed by a glass tube (2.5 cm × 8.0 cm high) containing a caterpillar. The assays were repeated twice on different days. The evaluations were carried out 24, 48, 72, 96, 120, 144, and 168 h after the application of the treatments, counting the number of live and dead insects. Insect survival, over time, was used to obtain time-response curves, and data, after 24 h of treatment, were used to estimate the dose-response (Section 2.5).

### 2.3.4. Chronic Toxicity of EOs against the FAW in an Ingestion Test

The EOs from *D. lanceolata* (leaves and stem barks) and *X. brasiliensis* (stem barks) were selected because of their acute toxicity in the topical application assay (Section 2.3.2). The EOs (200 mg) were dissolved in 20 mL of an aqueous 0.01 g mL$^{-1}$ Tween 80® solution. The resulting solution (20 mL) was incorporated into an artificial diet (200 mL). Thus, the EOs were offered to caterpillars in the concentration of 1 mg of EO/mL of diet. Pieces of diet (1.0 cm × 1.5 cm high) containing the treatments were transferred to glass tubes (2.5 cm × 8.0 cm high). Then, one FAW caterpillar (48-hour-old and previously maintained on an artificial diet) was transferred into each tube.

The assay was conducted in a completely randomized design with 50 repetitions per treatment, each formed by a tube containing a caterpillar. Negative controls consisted of diet plus water and aqueous 0.01 g mL$^{-1}$ Tween 80® solution (20 mL). The experiment was repeated twice on different days. Insect survival was evaluated 24, 48, 72, 96, 120, 144, and 168 h after offering the diets containing the EOs to the caterpillars. After 168 h of providing the diet with the EOs for the FAW, the mass of live caterpillars was measured.

#### 2.3.5. Food Preference Test of the FAW by EOs

The EOs of *D. lanceolata* (leaves and stem barks) were selected for causing a reduction in the mass of *S. frugiperda* caterpillars in the chronic toxicity test by ingestion (Section 2.3.4). For that, the EOs (100 mg) were dissolved in an aqueous 0.01 g mL$^{-1}$ Tween 80$^®$ solution (10 mL) and incorporated into the artificial diet (100 mL). Pieces of diet (1.5 cm in diameter × 1.3 cm in height), previously weighed, were placed equidistantly in an arena consisting of a Petri dish (15 cm in diameter × 1.9 cm in height). Two pieces of diet were placed in each arena, one containing the treatment with the EO and the other with the diet containing aqueous 0.01 g mL$^{-1}$ Tween 80$^®$ solution. In the control treatment arena, a piece of diet was used in which distilled water was added, and another with an aqueous 0.01 g mL$^{-1}$ Tween 80$^®$ solution on it. In the center of each arena, five seven-day-old *S. frugiperda* caterpillars were released (Figure 1). They had been previously kept without food for 2 h before being released in the center of each arena. The diet's aliquots (1.0 cm in diameter × 1.5 cm in height) were used to determine the initial dry mass.

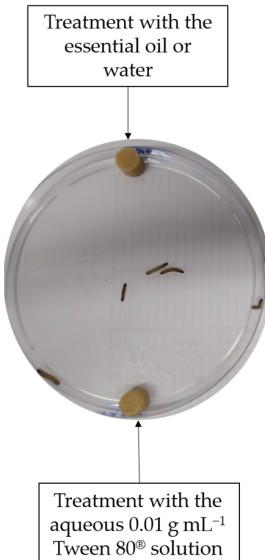

**Figure 1.** Experimental arena of food preference test of the *Spodoptera frugiperda* by essential oils.

The bioassay was performed in a completely randomized design with 50 replications per treatment, each consisting of a Petri dish with five caterpillars. The non-preference evaluation of the caterpillars for the EO-containing diet was calculated as a function of the percentage of caterpillars present in each treatment 24, 48, and 72 h after insect release. Only those caterpillars on a diet at the time of evaluation were counted.

Diet pieces not provided to the insect were dried in an oven at 45 °C for 24 h to determine the initial dry mass of the diet. Seventy-two hours after the caterpillars' release into the arenas, the unconsumed diet was submitted to drying in the same conditions to determine the dry mass of the consumed diet. Food consumption corresponded to the initial diet dry mass (g)-the dry mass of the remaining diet pieces (g).

#### 2.3.6. Toxicity of Major Compounds from *Duguetia lanceolata* (Leaves) EO against the FAW

For this bioassay, the major compounds of the EO from the leaves of *D. lanceolata*, β-caryophyllene (Sigma-Aldrich$^®$; purity ≥ 80%) and caryophyllene oxide (Sigma-Aldrich$^®$; purity ≥ 99%) (Table 2) were employed. This EO was selected because it showed the lowest LD$_{90}$ value for the FAW caterpillars in a topical application assay (Section 3.2.2). The treatments consisted of negative control (acetone); EO of *D. lanceolata* leaves (38.33 µg of EO/insect); β-caryophyllene (3.36 µg of compound/insect); caryophyllene oxide (3.6 µg of compound/insect) and the mixture of β-caryophyllene (3.36 µg of compound/insect) and caryophyllene oxide (3.6 µg of compound/insect).

**Table 2.** Results of the analysis by gas chromatography coupled to mass spectrometry, of essential oils from *Annona neolaurifolia* (leaves), *Duguetia lanceolata* (leaves and stem barks), and *Xylopia brasiliensis* (branches and stem barks).

| Compound | Peak Area [a] (%) | Probability [b] (%) | Molecular Formula | AI (Calculated) ** | AI (Literature) *** |
|---|---|---|---|---|---|
| | | | *Annona neolaurifolia* (leaves) | | |
| α-pinene | 6.80 | 97 | $C_{10}H_{16}$ | 931 | 932 |
| Sabinene | 1.47 | 94 | $C_{10}H_{16}$ | 970 | 969 |
| β-pinene | 6.35 | 97 | $C_{10}H_{16}$ | 974 | 974 |
| Linalool | 5.45 | 93 | $C_{10}H_{18}O$ | 1099 | 1095 |
| α-terpineol | 1.55 | 94 | $C_{10}H_{18}O$ | 1188 | 1186 |
| α-copaene | 1.57 | 94 | $C_{15}H_{24}$ | 1371 | 1374 |
| β-cubebene | 1.07 | 94 | $C_{15}H_{24}$ | 1386 | 1387 |
| (*E*)-caryophyllene | 13.70 | 94 | $C_{15}H_{24}$ | 1414 | 1417 |
| α-humulene | 2.93 | 95 | $C_{15}H_{24}$ | 1448 | 1452 |
| γ-muurolene | 1.44 | 92 | $C_{15}H_{24}$ | 1473 | 1478 |
| NI * | 5.94 | - | - | 1476 | - |
| NI * | 1.19 | - | - | 1490 | |
| δ-cadinene | 1.26 | 89 | $C_{15}H_{24}$ | 1520 | - |
| Spathulenol | 2.65 | 92 | $C_{15}H_{24}O$ | 1572 | 1577 |
| Caryophyllene oxide | 7.98 | 95 | $C_{15}H_{24}O$ | 1577 | 1582 |
| NI * | 1.00 | - | - | 1602 | - |
| | | | *Duguetia lanceolata* (leaves) | | |
| α-pinene | 1.80 | 97 | $C_{10}H_{16}$ | 931 | 932 |
| β-pinene | 3.38 | 98 | $C_{10}H_{16}$ | 975 | 974 |
| β-caryophyllene | 7.43 | 94 | $C_{15}H_{24}$ | 1414 | 1417 |
| α-caryophyllene | 1.11 | 95 | $C_{15}H_{24}$ | 1448 | 1452 |
| NI * | 1.28 | - | - | 1467 | - |
| NI * | 1.47 | - | - | 1476 | - |
| NI * | 1.02 | - | - | 1483 | - |
| NI * | 2.48 | - | - | 1521 | - |
| NI * | 4.07 | - | - | 1550 | - |
| Spathulenol | 4.59 | 92 | $C_{15}H_{24}O$ | 1572 | 1577 |
| Caryophyllene oxide | 9.42 | 94 | $C_{15}H_{24}O$ | 1577 | 1582 |
| Elemol | 1.16 | 91 | $C_{15}H_{26}O$ | 1587 | 1548 |
| Guaiol | 2.27 | 92 | $C_{15}H_{26}O$ | 1594 | 1600 |
| NI * | 3.92 | - | - | 1608 | - |
| NI * | 1.53 | - | - | 1624 | - |
| NI * | 1.40 | - | - | 1634 | - |
| NI * | 1.78 | - | - | 1636 | - |
| NI * | 4.44 | - | - | 1644 | - |
| NI * | 1.63 | - | - | 1652 | - |
| NI * | 1.08 | - | - | 1664 | - |
| NI * | 1.81 | - | - | 1682 | - |
| NI * | 1.03 | - | - | 1757 | - |
| NI * | 1.75 | - | - | 1773 | - |
| NI * | 1.19 | - | - | 1783 | - |
| NI * | 1.32 | - | - | 1897 | - |
| | | | *Duguetia lanceolata* (stem bark) | | |
| α-pinene | 9.70 | 97 | $C_{10}H_{16}$ | 931 | 932 |
| β-pinene | 13.55 | 97 | $C_{10}H_{16}$ | 975 | 974 |
| α-Cubebene | 1.15 | 95 | $C_{15}H_{24}$ | 1346 | 1348 |
| NI * | 2.66 | - | - | | - |
| γ-muurolene | 1.25 | 92 | $C_{15}H_{24}$ | 1472 | 1479 |
| δ-cadinene | 1.55 | 90 | $C_{15}H_{24}$ | 1520 | 1522 |
| Spathulenol | 3.18 | 92 | $C_{15}H_{24}O$ | 1572 | 1577 |
| Caryophyllene oxide | 1.80 | 94 | $C_{15}H_{24}O$ | 1576 | 1582 |
| Guaiol | 1.01 | 91 | $C_{15}H_{26}O$ | 1593 | 1600 |
| NI * | 2.46 | - | - | | - |
| NI * | 4.15 | - | - | | - |
| NI * | 1.94 | - | - | 1637 | 1645 |
| NI * | 6.36 | - | - | | - |
| NI * | 12.47 | - | - | | - |
| NI * | 1.81 | - | - | | - |
| NI * | 1.18 | - | - | | - |
| NI * | 1.82 | - | - | | - |
| NI * | 1.16 | - | - | | - |
| NI * | 1.24 | - | - | | - |

**Table 2.** *Cont.*

| Compound | Peak Area [a] (%) | Probability [b] (%) | Molecular Formula | AI (Calculated) ** | AI (Literature) *** |
|---|---|---|---|---|---|
| | | *Xylopia brasiliensis* (branches) | | | |
| α-pinene | 1.10 | 97 | $C_{10}H_{16}$ | 931 | 932 |
| Camphene | 1.60 | 97 | $C_{10}H_{16}$ | 945 | 946 |
| β-pinene | 1.00 | 97 | $C_{15}H_{24}$ | 974 | 974 |
| Eucalyptol | 2.43 | 95 | $C_{10}H_{18}O$ | 1028 | 1026 |
| Nopinone | 1.47 | 90 | $C_9H_{14}O$ | 1135 | 1135 |
| Myrtenal | 1.73 | 95 | $C_{10}H_{14}O$ | 1194 | 1195 |
| Verbenone | 1.44 | 94 | $C_{10}H_{14}O$ | 1206 | 1204 |
| β-elemene | 1.57 | 96 | $C_{15}H_{24}$ | 1388 | 1389 |
| β-caryophyllene | 1.58 | 94 | $C_{15}H_{24}$ | 1414 | 1417 |
| Spathulenol | 43.14 | 94 | $C_{15}H_{24}O$ | 1576 | 1577 |
| NI * | 7.92 | - | - | 1579 | 1582 |
| NI * | 5.26 | - | - | 1626 | - |
| NI * | 2.45 | - | - | 1635 | - |
| NI * | 1.18 | - | - | 1639 | - |
| NI * | 1.61 | - | - | 1650 | - |
| NI * | 1.03 | - | - | 1674 | - |
| | | *Xylopia brasiliensis* (stem bark) | | | |
| α-pinene | 5.70 | 97 | $C_{10}H_{16}$ | 931 | 932 |
| Camphene | 6.10 | 97 | $C_{10}H_{16}$ | 946 | 946 |
| β-pinene | 7.00 | 98 | $C_{10}H_{16}$ | 975 | 974 |
| *Trans*-pinocarveol/Pinocarveol | 1.03 | 95 | $C_{10}H_{16}O$ | 1135 | 1135 |
| Myrtenal | 1.03 | 95 | $C_{10}H_{14}O$ | 1194 | 1195 |
| NI * | 3.19 | - | - | 1372 | - |
| NI * | 2.59 | - | - | 1440 | - |
| NI * | 1.21 | - | - | 1473 | - |
| α-curcumene | 1.60 | 93 | $C_{15}H_{22}$ | 1480 | 1479 |
| NI * | 1.34 | - | - | 1509 | - |
| NI * | 1.98 | - | - | 1520 | - |
| NI * | 1.75 | - | - | 1550 | - |
| NI * | 1.35 | - | - | 1561 | - |
| Spathulenol | 7.94 | 94 | $C_{15}H_{24}O$ | 1573 | 1577 |
| Caryophyllene oxide | 5.24 | 89 | $C_{15}H_{24}O$ | 1577 | 1582 |
| NI * | 1.40 | - | - | 1587 | - |
| NI * | 1.28 | - | - | - | - |
| NI * | 2.21 | - | - | 1607 | - |
| NI * | 2.27 | - | - | 1623 | - |
| NI * | 1.54 | - | - | 1627 | - |
| Cubenol | 2.82 | 85 | $C_{15}H_{26}O$ | 1638 | 1645 |
| NI * | 1.83 | - | - | 1602 | - |
| NI * | 2.44 | - | - | 1607 | 1652 |
| β-bisabolol | 4.51 | 83 | $C_{15}H_{26}O$ | 1623 | 1674 |
| NI * | 4.67 | - | - | 1627 | - |
| NI * | 2.26 | - | - | 1638 | - |
| NI * | 1.06 | - | - | 1643 | - |
| NI * | 1.22 | - | - | 1648 | 1579 |

[a] Peak area = (100 × area of the peak in the chromatogram)/($\sum$ areas of all peaks in the chromatogram); [b] Probability = probability calculated by the GC-MS software that the corresponding mass spectrum originates from the substance described in the first column of this table; * NI = Not identified; ** AI = calculated arithmetic index; *** AI = arithmetic index according to [28].

The dose was determined according to the $LD_{50}$ obtained for EOs (Section 3.2.2). For the calculations, the following formula was used:

$$\text{Dose to be tested} = \text{dose of EO}\left(LD_{50)}\right) \times \text{content of the main compound } (\%) \qquad (2)$$

A correction factor was also used according to the purity of the main compound described by the manufacturer. The assay was conducted as described in Section 2.3.2.

*2.4. Selectivity of EOs for Trichogramma pretiosum*

2.4.1. General Procedures

The parasitoid *T. pretiosum* and unviable eggs of the alternative host, *Ephestia kuehniella* (Zeller, 1879) (Lepidoptera: Pyralidae), were acquired from the company Promip-Manejo Integrado de Pragas (Engenheiro Coelho–São Paulo, Brazil).

The EOs from the leaves (70.76 µg of EO/µL of acetone) and stem bark (127.15 µg of EO/µL of acetone) of *D. lanceolata*, which caused high mortality for *S. frugiperda* caterpillars (Sections 3.2.1 and 3.2.2), were used in concentrations equivalent to those estimated of the LD$_{90}$ values for *S. frugiperda*.

The assay was conducted in a climatic chamber at a temperature of $25 \pm 2$ °C, relative humidity of $70 \pm 10$%, and a 12L:12D photoperiod.

2.4.2. Selectivity of EOs on the Immature Phases of *Trichogramma pretiosum*

Per treatment, 30 females of *T. pretiosum* were individualized in glass tubes (2.5 cm in diameter $\times$ 8.0 cm in height) and fed with drops of pure honey. Approximately 125 *E. kuehniella* eggs were adhered to blue cardboard sheets (5 cm high $\times$ 0.5 cm wide) using gum Arabic diluted in water (1:1, *v/v*) and offered to each female of the parasitoids for 24 h. Then, the females were discarded, and the supposedly parasitized eggs were used to conduct the bioassay according to each desired developmental stage of the parasitoid.

Cards with *E. kuehniella* eggs (30 per treatment) containing the parasitoid in the egg-larvae period (0–24 h after parasitism), pre-pupal phase (72–96 h after parasitism), and pupa (168–192 h after parasitism) [31] were immersed in the EOs solutions, previously dissolved in acetone, for five seconds, according to the method described in the literature [32,33]. After evaporating the solvent in a hood for 1 h, the cards were placed in new tubes and maintained in an acclimatized chamber.

A completely randomized experimental design was used, with 30 repetitions, each formed by a tube containing a card with *E. kuehniella* eggs containing *T. pretiosum* in the egg-larva period or the pre-pupa or pupa stages inside. The effects of the EOs on the F1 generation parasitoids were evaluated as a function of the emergence percentage and according to the sex ratio of the parasitoids when treated at different development stages.

2.4.3. Selectivity of EOs for *Trichogramma pretiosum* Adults

Thirty *T. pretiosum* females per treatment were individualized in glass tubes and fed with pure honey in the form of a droplet deposited on the wall of each tube. About 125 eggs of *E. kuehniella* adhered to blue cardboard sheets (5 cm high $\times$ 0.5 cm wide) with gum arabic diluted in water (1:1, *v/v*). Then, the cards containing the eggs were treated with the EOs described in the literature [32,33]. After 1 h in a hood to eliminate the solvent, the eggs were offered to the females for 24 h, for parasitism. Then, the females were kept in the same tubes to assess their longevity. At the same time, the cards containing the supposedly parasitized eggs were transferred to new tubes and kept in a climatized chamber under the same conditions described above until the parasitoids of the next generation F1 emerged.

The bioassay was conducted in a completely randomized design, with 30 repetitions, each formed by a tube containing a card with about 125 *E. kuehniella* eggs. The longevity of females, parasitism capacity of females of the F0 generation, and the emergence of insects of the F1 generation were evaluated.

2.4.4. Classification of EOs According to the International Organization for Biological Control

The classification of the selectivity of the compounds was performed following the International Organization for Biological Control (IOBC) criteria, where class I: harmless (parasitism reduction (RP) or emergence (RE) < 30%); class II: slightly harmful (30% $\leq$ (RP-RE) $\leq$ 79%); class III: moderately harmful (80% $\leq$ (RP-RE) $\leq$ 99%) and class IV: harmful (RP-RE) > 99%) [34,35].

### 2.5. Statistical Analysis

Data concerning the survival of insects over time (Sections 2.3.2–2.3.4 and Section 2.3.6) and longevity of *T. pretiosum* (Section 2.4.3) were submitted to survival analysis using the non-parametric test of Kaplan–Meier. The Pairwise multiple comparison test was utilized to compare the curves. The median lethal time ($LT_{50}$) for each treatment was estimated.

After 24 h of treatment application, data from insect mortality were submitted to logit analysis to determine the dose response (Section 2.3.3). The median lethal dose ($LD_{50}$) and lethal dose to 90% ($LD_{90}$) of the population were calculated using the following equation:

$$f(x) = 1/1 + \exp(b(\log(x) - \log(e))) \tag{3}$$

where "b" and "e" = coefficients of the equation.

Data related to caterpillar mass (Section 2.3.4), percentage of parasitoid emergence, and the number of parasitized eggs (Section 2.4) had normality and homoscedasticity verified using the Shapiro-Wilk and Bartlett tests. Data that did not follow a normal distribution were analyzed through non-parametric tests such as the Kruskal-Wallis test.

Data collected in the free-choice food preference test (Section 2.3.5) and related to sex ratio (Section 2.4) were analyzed using the chi-square test ($\chi^2$).

All analyzes were performed using the R software [36].

### 3. Results

#### 3.1. Chemical Characterization of EOs

The compound in greater quantity in the leaves of *A. neolaurifolia* was (*E*)-caryophyllene (13.7%). β-caryophyllene (7.43%) and caryophyllene oxide (9.42%) were the major compounds in the leaves of *D. lanceolata*. The major compound in the EO of the stem bark from *D. lanceolata* was β-pinene (13.55%). Spathulenol was the principal compound of the EOs from *X. brasiliensis* branches (41.14%) and stem barks (7.94%) (Table 2).

#### 3.2. Bioassays with the FAW

3.2.1. Acute Toxicity of EOs against the FAW in a Topical Application Test

The EOs from *A. neolaurifolia* (leaves), *D. lanceolata* (stem barks and leaves), and *X. brasiliensis* (stem barks) ($\chi^2$ = 472; df = 5; $p \leq 0.001$) showed toxicity to the FAW caterpillars, with survival probabilities equal to 8.3, 1.9, 0, and 0%, respectively. The $LT_{50}$ (time required to cause mortality of 50% in the population) was less than or equal to 24 h. Only the EO from the branches of *X. brasiliensis*, with a survival probability of 80.9%, presented extremely low activity against the FAW. The caterpillars treated with the negative control, acetone, showed a survival probability of 96% and $LT_{50}$ of more than 168 h (Figure 2).

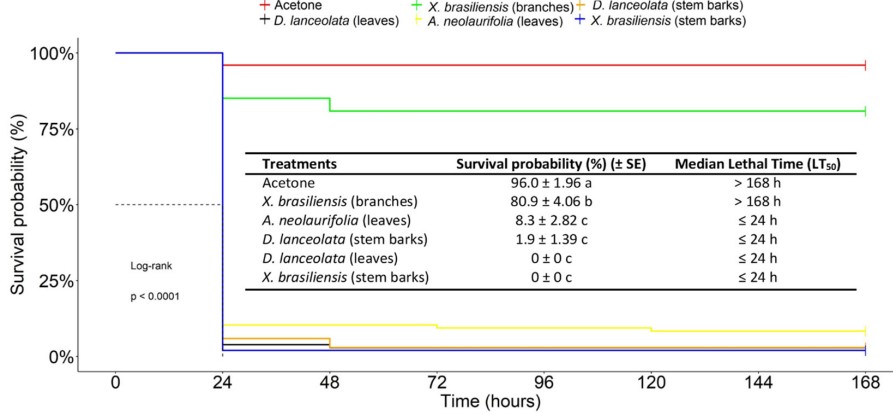

**Figure 2.** Survival analysis of *Spodoptera frugiperda* caterpillars subjected to topical application of essential oils of *Annona neolaurilofia* (leaves), *Duguetia lanceolata* (leaves and stem barks) and *Xylopia brasiliensis* (branches and stem barks) in the dose of 100 μg of EO/insect. The same letters do not differ statistically by Pairwise multiple comparison test.

### 3.2.2. Dose-Response and Time-Response Curves

The dose of EO from *D. lanceolata* (leaves) necessary to cause 90% mortality in the population of the FAW after 24 h of treatment application was 44% lower when compared to the EO from *D. lanceolata* (stem barks). This value was 500 times higher than that of the active ingredient cypermethrin. However, it must be considered that cypermethrin is a pure substance. In addition, pyrethroids are known to be used at low doses in the field (Table 3).

**Table 3.** Dose-mortality response of caterpillars of *Spodoptera frugiperda* treated with essential oils from the leaves and stem barks of *Duguetia lanceolata* and the active ingredient cypermethrin.

| Treatment | df | $\chi^2$ | $p$ | b * | e * | $LD_{50}$ (μg of EO/Insect) | $LD_{90}$ (μg of EO/Insect) |
|---|---|---|---|---|---|---|---|
| *Duguetia lanceolata* (stem bark) | 500 | 543.79 | 0.0858 | −1.34 | 24.75 | 24.75 ± 2.0589 | 127.14 ± 27.4170 |
| *Duguetia lanceolata* (leaves) | 498 | 477.05 | 0.7428 | −3.58 | 38.33 | 38.33 ± 1.3423 | 70.76 ± 4.3037 |
| Cypermethrin | 498 | 571.11 | 0.0514 | −0.83 | 0.01 | 0.01 ± 0.0019 | 0.14 ± 0.0421 |

* "b" and "e" = coefficients of the Equation (3).

It was observed that the EOs in higher doses had the lowest survival means. The EO from *D. lanceolata* (stem barks) ($\chi^2$ = 268; df = 5; $p \leq 0.001$) in doses 42, 64, and 100 μg of EO/insect caused death rates of 61.1, 86.6 and 96.1%, respectively. The $LT_{50}$ values for these doses were ≤24 h (Figure 3a). For the EO of the leaves of *D. lanceolata* ($\chi^2$ = 301; df = 5; $p \leq 0.001$), the doses of 28 and 50 μg of EO/insect caused death rates of 47.1 and 91.5%, respectively. Caterpillars treated with the EO from the leaves of *D. lanceolata* in the dose of 50 μg of EO/insect presented $LT_{50} \leq 24$ h (Figure 3b). As the active ingredient cypermethrin ($\chi^2$ = 443; df = 5; $p \leq 0.001$) allowed for less than 30% survival at doses greater than 0.032 μg of EO/insect, the corresponding $LT_{50}$ was ≤24 h (Figure 3c).

**a)**

| Treatments | Survival probability (%) (± SE) | Median Lethal (LT_{50}) |
|---|---|---|
| 0 μg of EO/larvae | 90.9 ± 2.89 a | > 168 h |
| 18 μg of EO/larvae | 86.1 ± 3.44 a | > 168 h |
| 28 μg of EO/larvae | 71.0 ± 4.54 b | > 168 h |
| 42 μg of EO/larvae | 38.4 ± 4.89 c | ≤ 24 h |
| 64 μg of EO/larvae | 13.4 ± 3.46 d | ≤ 24 h |
| 100 μg of EO/larvae | 3.9 ± 1.92 d | ≤ 24 h |

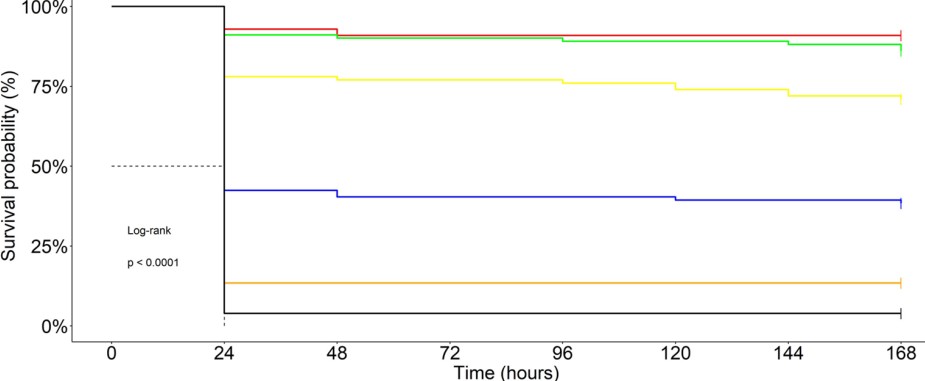

**Figure 3.** *Cont*.

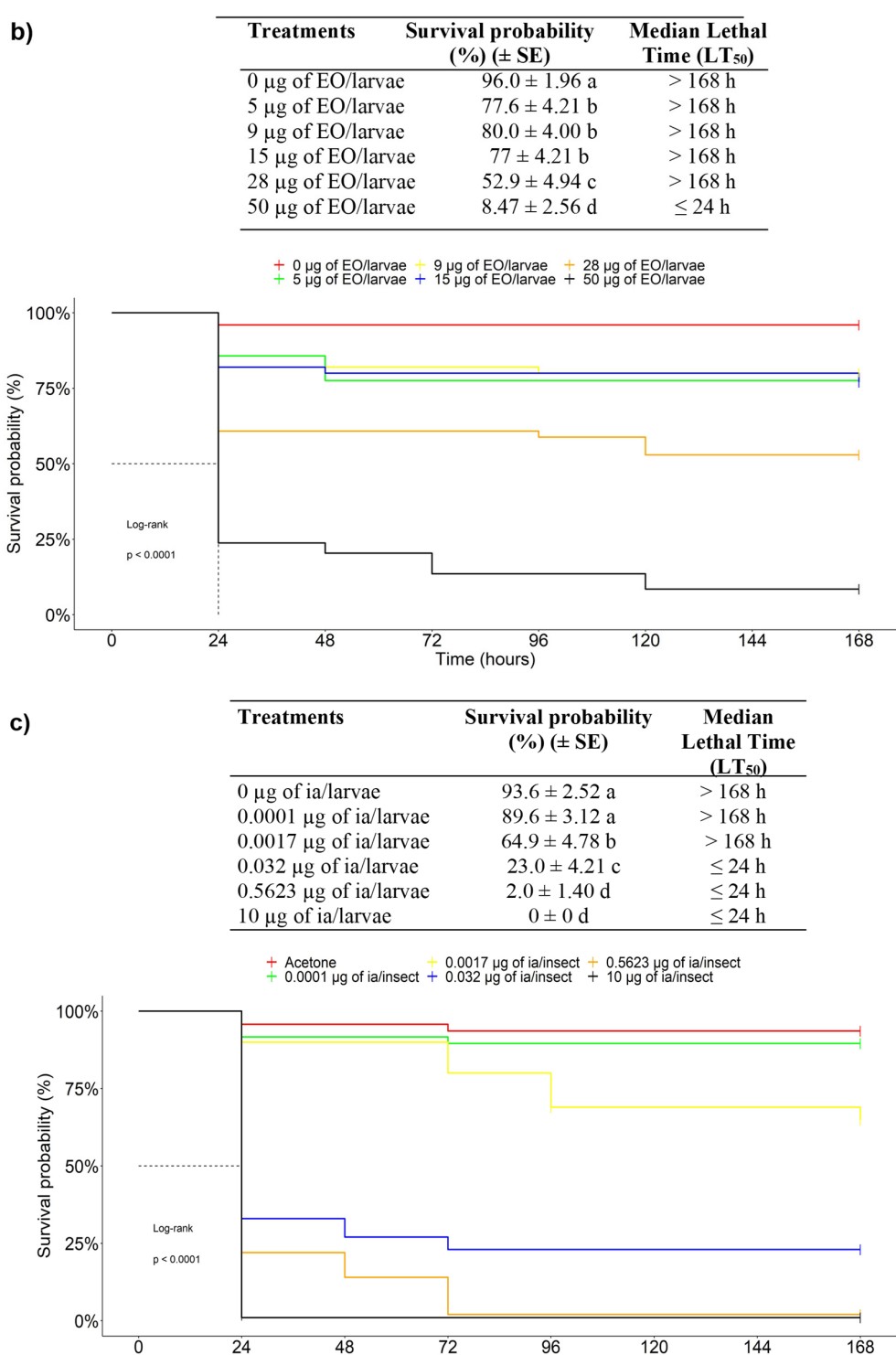

**b)**

| Treatments | Survival probability (%) (± SE) | Median Lethal Time (LT$_{50}$) |
|---|---|---|
| 0 µg of EO/larvae | 96.0 ± 1.96 a | > 168 h |
| 5 µg of EO/larvae | 77.6 ± 4.21 b | > 168 h |
| 9 µg of EO/larvae | 80.0 ± 4.00 b | > 168 h |
| 15 µg of EO/larvae | 77 ± 4.21 b | > 168 h |
| 28 µg of EO/larvae | 52.9 ± 4.94 c | > 168 h |
| 50 µg of EO/larvae | 8.47 ± 2.56 d | ≤ 24 h |

**c)**

| Treatments | Survival probability (%) (± SE) | Median Lethal Time (LT$_{50}$) |
|---|---|---|
| 0 µg of ia/larvae | 93.6 ± 2.52 a | > 168 h |
| 0.0001 µg of ia/larvae | 89.6 ± 3.12 a | > 168 h |
| 0.0017 µg of ia/larvae | 64.9 ± 4.78 b | > 168 h |
| 0.032 µg of ia/larvae | 23.0 ± 4.21 c | ≤ 24 h |
| 0.5623 µg of ia/larvae | 2.0 ± 1.40 d | ≤ 24 h |
| 10 µg of ia/larvae | 0 ± 0 d | ≤ 24 h |

**Figure 3.** Survival analysis of caterpillars of *Spodoptera frugiperda* subjected to topical application of essential oil of *Duguetia lanceolata* (stem barks) (**a**); *Duguetia lanceolata* (leaves) (**b**); and the active ingredient cypermethrin (**c**). The same letters do not differ statistically by Pairwise multiple comparison test.

### 3.2.3. Chronic Toxicity of EOs against the FAW in an Ingestion Test

Despite the insecticidal activity of the EOs from *D. lanceolata* (leaves and stem barks) in the acute toxicity test (Sections 3.2.1 and 3.2.2), the same effect was not observed when the FAW caterpillars were submitted to a chronic toxicity assay in an ingestion test ($\chi^2$ = 15.1; df = 4; $p$ = 0.005). The survival probability ranged from 73.5 to 90%, and $LT_{50}$ was >168 h for all treatments (Figure 4 ).

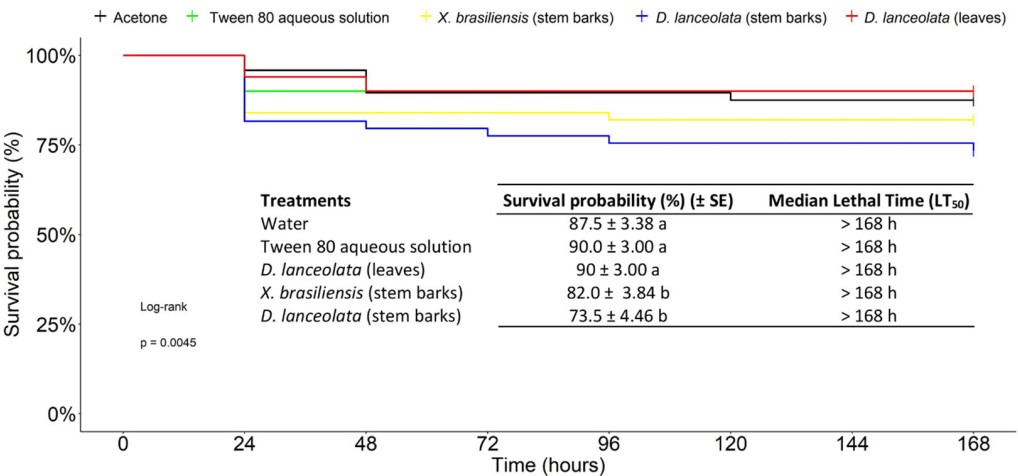

**Figure 4.** Survival analysis of *Spodoptera frugiperda* caterpillars offered an artificial diet containing essential oils from *Duguetia lanceolata* (leaves and stem barks) and *Xylopia brasiliensis* (stem bark). The same letters do not differ statistically by Pairwise multiple comparison test.

Although no lethal effect was found, all EOs caused a sublethal impact, such as a reduction in insect mass. The most promising results were found for the EOs from *D. lanceolata* (leaves and stem barks). *Spodoptera frugiperda* caterpillars fed on an artificial diet containing those oils showed a mass 1.88 times lower than that observed for the negative controls, which consisted of diet plus water and aqueous 0.01 g mL$^{-1}$ Tween 80$^{\circledR}$ solution ($\chi^2$ = 103.72, df = 4, $p \leq$ 0.001) (Figure 5).

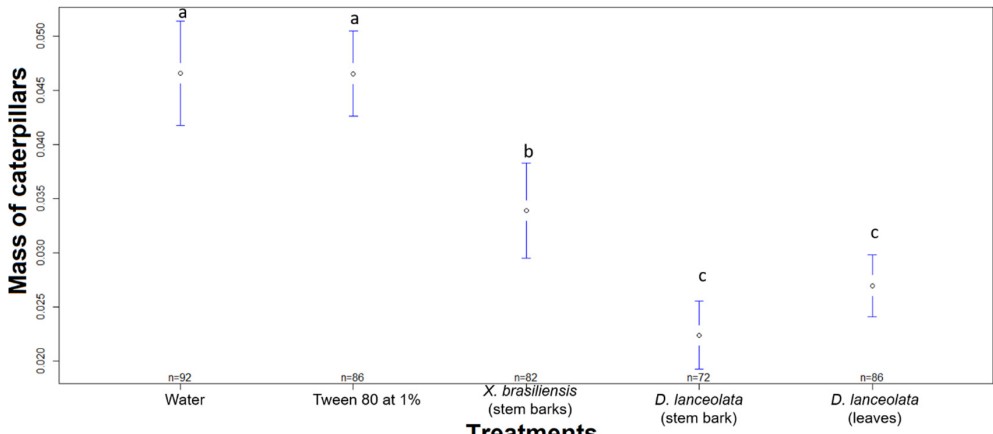

**Figure 5.** Mass of *Spodoptera frugiperda* caterpillars offered an artificial diet containing essential oils of *Duguetia lanceolata* (leaves and stem barks) and *Xylopia brasiliensis* (stem bark). The same letters do not differ statistically by the test of Kruskal-Wallis.

### 3.2.4. Food Preference Test of the FAW by EOs

The insects had no food preference (Figure S1) because the number of caterpillars on the pieces of the EO-containing diets did not differ from the number found in the control at the evaluated times. This indicates that the treatments do not have a repellent

or attractant effect on the insects. However, a 35.36% reduction in food consumption was observed for the treatment containing EOs from the stem bark of *D. lanceolata* (Figure 6), though it is insufficient to explain the insect mass reduction observed in the assay of chronic toxicity of EOs against *S. frugiperda* (Section 3.2.3). It is also noteworthy that the EOs of the leaves of *D. lanceolata* caused a reduction in insect mass (Section 3.2.3), though there was no reduction in consumption (Figure 6).

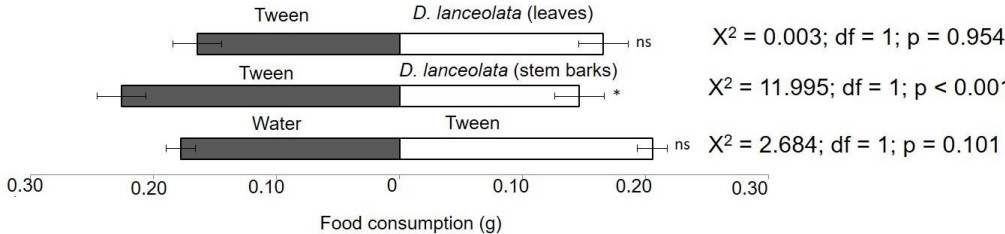

**Figure 6.** Food consumption (g) of *Spodoptera frugiperda* caterpillars exposed to diets treated with essential oils from the leaves and bark of the *Duguetia lanceolata* in food preference assay. The same letters do not differ statistically by $X^2$ test.

3.2.5. Toxicity of Major Compounds from *Duguetia lanceolata* (Leaves) EO against the FAW

When assessed separately, the compounds β-cariophyllene and caryophyllene oxide did not show insecticidal activity against FAW caterpillars. The EO from *D. lanceolata* caused a 44.1% reduction in the survival probability of insects. In comparison, the mixture formed by the β-caryophyllene and caryophyllene oxide caused a 27.3% reduction in the survival probability of *S. frugiperda*. Thus, although the results obtained with the mix of β-caryophyllene and caryophyllene oxide were not as toxic as the EOs, yet showed some toxicity to *S. frugiperda* ($\chi^2 = 55.9$; df = 4; $p \leq 0.001$) (Figure 7).

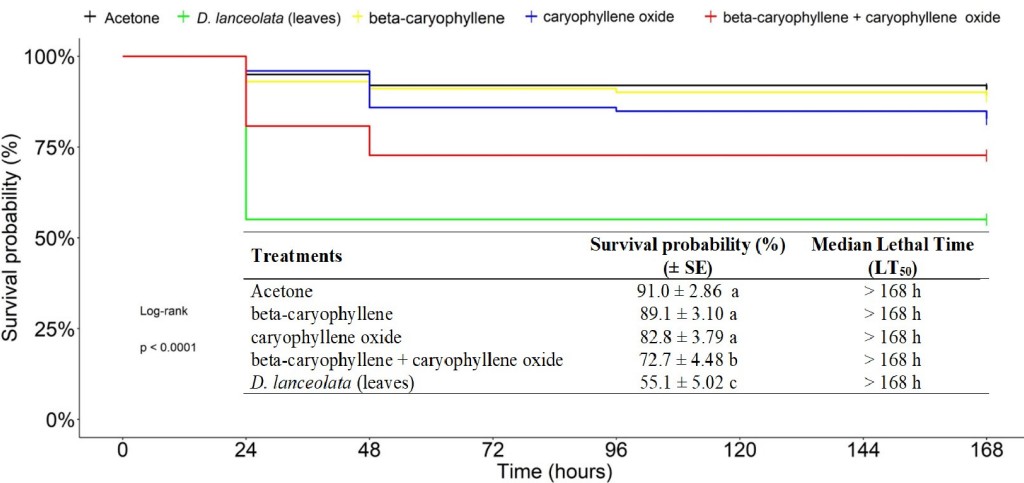

**Figure 7.** Survival analysis *Spodoptera frugiperda* larvae offered an artificial diet containing the essential oil of *Duguetia lanceolata* (leaves); β-caryophyllene; caryophyllene oxide; a mixture of β-caryophyllene and caryophyllene oxide; or the negative control: acetone. The same letters do not differ statistically by Pairwise multiple comparison test.

*3.3. Selectivity of EOs for Trichogramma pretiosum*

3.3.1. Selectivity of EOs on the Immature Phases of *Trichogramma pretiosum*

There was a difference in the percentage of the emergence of parasitoids that received treatments in the egg-larvae period ($\chi^2 = 52.66$; df = 2; $p < 0.01$), pre-pupal stage ($\chi^2 = 69.99$; df = 2; $p < 0.01$) and pupal stage ($\chi^2 = 66.65$; df = 2; $p < 0.01$). The EO of *D. lanceolata* (stem barks) was classified into toxicity classes III, IV, and III for egg-larvae, pre-pupal and pupal stages, respectively. In comparison, the EO from the leaves of *D. lanceolata* was classified in

the classes I, II, and III for egg-larvae, pre-pupal and pupal stages, respectively. Regarding the sex ratio, there was no statistical difference when the EO of *D. lanceolata* (leaves) was applied to the alternate host containing *T. pretiosum* in the stages of egg-larvae ($\chi^2 = 3.39$; df = 1; $p = 0.0653$) and pre-pupa ($\chi^2 = 1.27$; df = 1; $p = 0.2589$) (Table 4).

**Table 4.** Emergence (%), emergence reduction (ER), toxicological class (TC), and sex ratio (SR) for *Trichogramma pretiosum* adults treated with essential oils from the stem barks and leaves of *Duguetia lanceolata* in the egg-larvae, pre-pupal and pupal stages inside the eggs of *Ephestia kuehniella*.

| Treatment | Emergence (%) | ER (%) | TC * | SR | Emergence (%) | ER (%) | TC * | SR | Emergence (%) | ER (%) | TC * | SR |
|---|---|---|---|---|---|---|---|---|---|---|---|---|
| Acetone | 80.7 ± 5.02 a | - | - | 0.54 ns | 80.2 ± 3.89 a | - | - | 0.63 ± 0.07 ns | 78.70 ± 4.39 a | - | - | 0.53 ± 0.03 |
| Essential oil from *D. lanceolata* (stem bark) | 5.0 ± 3.7 b | 93.8 | III | ** | 0.0 ± 0.00 c | 100 | IV | | 4.00 ± 4.00 b | 94.9 | III | ** |
| Essential oil from *D. lanceolata* (leaves) | 68.3 ± 6.4 a | 14.9 | I | 0.40 ns | 24.2 ± 3.46 b | 69.8 | II | 0.45 ± 0.10 ns | 3.71 ± 2.03 b | 95.3 | III | ** |

* Toxicological class (TC) according to IOBC where class I: harmless (emergence reduction (ER) < 30%); class II: slightly harmful (30% ≤ ER ≤ 79%); class III: moderately harmful (80% ≤ ER ≤ 99%) and class IV: harmful (ER > 99%) [34]; ** It was not evaluated due to a large number of deaths. Mean followed by same letters, in the columns, do not differ statistically by Kruscal-Wallis test.

### 3.3.2. Selectivity of EOs for *Trichogramma pretiosum* Adults

The EOs from *D. lanceolata* (leaves and stem barks) did not show acute toxicity to adult females of *T. pretiosum* ($\chi^2 = 43.6$; df = 2; $p = 0.0001$). However, there was a reduction in the survival probability of the insect over time. The mortality peak for the treatment with the EO of *D. lanceolata* (stem barks) was observed after 48 h of insect exposure to the treatment; at the end of the assay, the probability of survival was 0%. For the EOs from *D. lanceolata* leaves, the survival probability at the end of the test was 6.7%, while that of the negative control, acetone, was 10%. The $LT_{50}$ values were: 264, 180, and 48 h for acetone, *D. lanceolata* (leaves), and *D. lanceolata* (stem barks), respectively (Figure 8).

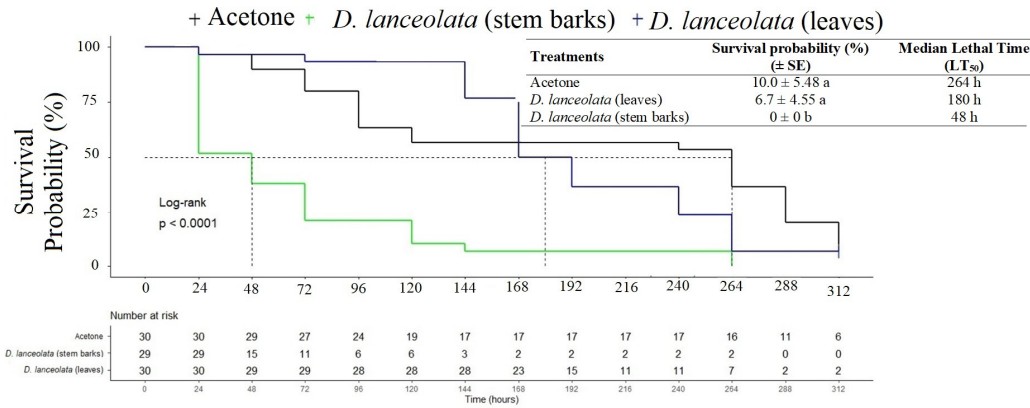

**Figure 8.** Survival analysis of adult females of *Trichogramma pretiosum* that received eggs from *Ephestia kuehniella* treated with essential oils from the leaves and stem barks of *Duguetia lanceolata*. The same letters do not differ statistically by Pairwise multiple comparison test.

The adult females of *T. pretiosum*, which received eggs from *E. kuehniella* treated with the EOs of stem barks and leaves from *D. lanceolata*, did not parasitize the eggs of the alternate host ($\chi^2 = 82.666$; df = 2; $p < 0.01$). Therefore, both oils are classified in class IV of the IOBC (Table 5).

**Table 5.** Number of eggs and reduction percentage parasitized by females of *Trichogramma pretiosum* when exposed to eggs of *Ephestia kuehniella* treated with essential oils from the leaves and stem barks of *Duguetia lanceolata*.

| Treatment | Number of Parasitized Eggs | ER (%) | TC * |
|---|---|---|---|
| Acetone | 27.40 ± 1.94 a | - | - |
| Essential oil from *D. lanceolata* (stem bark) | 0.00 ± 0.00 b | 100.0 | IV |
| Essential oil from *D. lanceolata* (leaves) | 0.03 ± 0.03 b | 99.9 | IV |

* Toxicological class (TC) according to IOBC where class I: harmless (emergence reduction (ER) < 30%); class II: slightly harmful (30% ≤ ER ≤ 79%); class III: moderately harmful (80% ≤ ER ≤ 99%) and class IV: harmful (ER > 99%) [34]. The same letters do not differ statistically by Kruscal-Wallis test.

## 4. Discussion

In the present study, acute toxicity was observed in the EOs from *A. neolaurifolia* (leaves), *D. lanceolata* (stem bark and leaves), and *X. brasiliensis* (stem bark) against *S. frugiperda* following topical application (Figure 2). For chronic toxicity test of EOs against *S. frugiperda* in an ingestion test, the EOs of *D. lanceolata* (stem barks and leaves) and *X. brasiliensis* (stem barks) were selected because they induced acute toxicity to *S. frugiperda* and displayed higher extraction yields (Table 1). However, these treatments did not reduce insect survival (Figure 4). Still, they caused a sublethal effect to reduce the mass of the caterpillars, with the most promising results found for the EO of the leaves and stem bark of *D. lanceolata* (Figure 5). Thus, it can be hypothesized that these EOs also have a phagodeterrent effect. However, in the food preference free-choice test, non-preference was observed by the EOs from *D. lanceolata* (leaves and stem bark) (Figure S1), which demonstrates that there is no repellency effect, which could be a factor to explain the reduction in the mass of insects. When analyzing the food consumption of the caterpillars, in a free-choice test, only for the treatment with the EO of *D. lanceolata* (stem bark), there was a reduction in insect food consumption (Figure 6).

Considering that the topical application resulted in rapid death of the insects (Figure 2), it is possible that compounds toxic to *S. frugiperda*, present in the EOs of *D. lanceolata* (leaves and stem bark), act directly on the nervous system (NS) of these insects. When penetrating through the integument of *S. frugiperda* and reaching the hemolymph, the components of such EOs may have acted directly on NS targets, such as other secondary metabolites that can act, for example, on the acetylcholinesterase enzyme [37,38], or cause alteration in octopamine [39] and γ-aminobutyric acid receptors [40].

On the other hand, when the EOs were incorporated into the artificial diet, it is plausible that they were mainly metabolized in insects' guts, preventing the absorption of the substances. This possibility is in accordance with the reported literature indicating that in the fat body, the midgut and Malpighian tubules are the main organs of insects that metabolize xenobiotics [41]. It is known that in the lumen of the intestine of insects, a series of enzymes, such as monoxidases, esterases, hydrolases, and transferases, can function as reductants and contribute to the detoxification of xenobiotics [42–44]. One of the mechanisms to detoxify secondary lipolytic metabolites is their functionalization, which can take place through oxidation, for example, and then conjugation with a polar compound, converting the metabolite into a water-soluble product that can be easily excreted [45]. The vital role of the intestinal microbiota in the detoxification of both synthetic chemical insecticides and secondary metabolites can be mentioned. It has already been found that resistant strains of *S. frugiperda* are an excellent reservoir of insecticide-degrading bacteria, demonstrating that the resistance mechanism is involved with the composition of the intestinal microbiota [46]. Thus, the reduction in the insect's mass, without preference and reduction in food consumption, may have occurred due to an increase in energy consumption for xenobiotic detoxification [47]. Therefore, these results open perspectives

for developing new works aiming to elucidate the detoxification mechanisms of *S. frugiperda* to these compounds.

When subjected to topical EO application, the insects showed rapid death at low doses. Notably, at doses above 48 and 50 μg of EO/insect, the $LT_{50}$ was ≤24 h for the EOs of the stem bark and leaves of *D. lanceotata* (Figure 3), and the respective $LD_{50}$ values were 24.75 and 38.33 μg of EO/insect (Table 3). The results obtained after topical applications of the EOs on the insect are promising compared to other botanical insecticides reported in the literature. Pyrethrins, for example, were the third most used botanical insecticide in the state of California in 2016, with 3355 kg applied [48]. The mean values of $LD_{50}$ and $LD_{90}$ found for the EOs used in this work were respectively 4.63 and 4.89 bigger than that of pyrethrum extract consisting of a mixture of more than 50% of pyrethrins (50%, Sigma-Aldrich) for *S. littoralis* caterpillars [49]. Considering that a commercial product with a high degree of purity was used in the previous work, the good insecticidal activity of these EOs becomes even more evident.

The species *D. lanceolata* is a native plant of the Americas. Similar to other Annonaceae that have been drawing attention for their pesticidal activity, studies with this species have intensified in recent years. Moreover, the acaricidal activity against *Tetranychus* spp. (Acari: Tetranychidae) [50] and *Dermanyssus gallinae* (De Geer, 1778) (Mesostigmata: Dermanyssidae) [51]; toxicity to insects of medical importance, such as *Culex quinquefasciatus* (S.) (Diptera: Culicidae) [52] and for stored grain pests such as *Sitophilus zeamais* (Motschulsky) (Coleoptera: Curculionidae) and *Zabrotes subfasciatus* (Bohemann, 1833) (Coleoptera, Chrysomelidae, Bruchinae) [53–55]. In the specific case of the FAW, this is the first bioactivity study using this plant's EOs in a topical application test for this insect. Previous studies carried out by our research group verified the insecticidal activity of the fraction soluble in dichloromethane from the methanolic extract from stem barks of this plant against this insect in an ingestion test. Employing metabolomic analysis and nuclear magnetic resonance (NMR), the authors attributed the harmful activity to 2,4,5-trimethoxystyrene [20]. Subsequently, these results were confirmed by isolating *trans*-asarone and 2,4,5-trimethoxystyrene, which are active for the FAW. Previous work reported that 2,4,5-trimethoxystyrene showed more significant toxicity [21]. In another study, in which the ethanolic extract of the leaves of *D. lanceolata* was incorporated into the artificial diet of the FAW, it was found that it caused only sublethal effects on insects [18].

The differences between the results previously described in the literature, which demonstrate toxicity in an ingestion test when this plant extract was studied, and those reported in the present work using EOs, can be explained by the differences in extraction methods used. In addition, biotic and abiotic factors can cause changes in the profile of secondary metabolites. Studies that aim to evaluate the chemical profile of the EOs of *D. lanceolata* are still scarce in the literature. In this study, the EOs of the stem bark of *D. lanceolata* presented β-pinene (13.55%) as the major compound. It stands out that the caryophyllene oxide was also detected, but in a smaller quantity (1.80%). While on the leaves of *D. lanceolata*, β-caryophyllene (7.43%) and caryophyllene oxide (9.42%) were the compounds detected in more significant proportions. The presence of caryophyllene oxide in the EOs of *D. lanceolata* corroborates the chromatographic analysis of the EOs from leaves [56] and stem bark of this plant [52]. It is noteworthy that it is suggested in the literature that caryophyllene oxide is one of the compounds that function as a chemical marker of this species [56]. On the other hand, it was reported that β-bisabolene (56.2%), 2,4,5-trimethoxystyrene (8.6%), and β-caryophyllene (3.8%) are the major constituents of *D. lanceolata* leaf EOs [53].

The major compounds in the EO of *D. lanceolata* (leaves), β-caryophyllene, and caryophyllene oxide were not toxic against the FAW when evaluated separately (Figure 7). When the substances were combined and tested, the probability of insect survival was attenuated; however, this result was not as significant as observed with the EOs. Thus, there is a synergism or additive effect between these substances. Furthermore, the fact that the insecticidal activity was not as pronounced as that of the EO suggests that other substances

present in the EO may also influence the insecticidal activity of *S. frugiperda*. Therefore, it is necessary to use other chemical methods, such as NMR, to identify the different components of the EO and submit them to tests with *S. frugiperda*. Since the relationships between the chemical composition of EOs and the bioactivity of the compounds are not well understood, it can be mentioned that it has already been found that even substances that do not present insecticidal activity alone are necessary for the mixture to be active [57]. A study can be mentioned in which greater EO toxicity was found; however when the identified substances were artificially mixed and tested the bioactivity was reduced [58]. Notably, synergistic or additive effects are common in botanical insecticides [57–60] and can be considered desirable. It was considered that the existence of several substances that act at different sites of action could delay the selection of populations of resistant insects.

Concerning previous reports of the insecticidal activity of the major compounds studied in the present work, it can be mentioned that when studying the bioactivity of three species of *Piper* spp. against *S. frugiperda*, it was found that they had in common the β-caryophyllene [61]. β-caryophyllene and caryophyllene oxide have already been reported to cause high insecticidal activity for *S. frugiperda* caterpillars, in an ingestion test, with mortality rates ranging from 5 to 75%. However, these authors used a concentration range of 80 to 1000 μg/mL, which is much superior to the one used in the present work [62]. Similarly, the β-caryophyllene incorporated into the artificial diet (250 μg/g) caused changes in this insect's feeding and oviposition behavior [63]. In this sense, research suggests that β-caryophyllene is one of the volatile compounds involved in the resistance of a particular maize lineage to the FAW [64]. To our knowledge, the mode of action of β-caryophyllene and caryophyllene oxide for insects has not yet been determined.

Although the EO of stem barks from *X. brasiliensis* is toxic in a topical application test for *S. frugiperda* caterpillars, the same bioactivity was not contacted for the EO from the branches of this plant. This finding can be explained by the fact that plants vary the composition and concentration of secondary metabolites differently according to the plant organ, which will influence the biological activity [65,66]. In this work, the major compound of *X. brasiliensis* (branches) was spathulenol (43.14%). While for the EO of stem bark, the compounds found in more significant proportions were: spathulenol (7.94%), β-pinene (7.0%), camphene (6.1%), caryophyllene oxide (5.24%), and β-bisabolol (4.51%). This study is the first to report the insecticidal activity of the EO from *X. brasiliensis* against the FAW. Previous works conducted using the ethanol extract and soluble fraction in dichloromethane from the methanolic extract of *X. brasiliensis* did not find a pesticidal activity for *Sitophilus zeamais* (Motschulsky) (Coleoptera: Curculionidae) [67], *Tetranychus tumidus* Banks (Acari: Tetranychidae) [50] and *S. frugiperda* in an ingestion test [20]. Although there are few studies aimed at evaluating the chemical characterization of EOs from *X. brasiliensis*, it can be mentioned that spathulenol (40.8%) is the major component of the EO of the leaves of this plant [68].

Another species studied in this work that showed insecticidal activity against *S. frugiperda* was *A. neolaurifolia*. To the authors' best knowledge, no works analyzed the chemical characterization of the EOs of this species. However, a few studies in the literature with this species refer to the isolation of acetogenins from plant extracts [69–71]. Previous studies conducted with the dichloromethane-soluble fraction of the methanolic extract of this plant did not show insecticidal activity against *S. frugiperda* in the ingestion test. The compounds found in significant proportions in the EOs were: (*E*)-caryophyllene (13.70%), followed by caryophyllene oxide (7.98%).

The EO of stem bark from *D. lanceolata* was not selective for immature stages of *T. pretiosum*. Regarding the EO from the leaves of this plant, it was more selective for the egg larvae than the pre-pupae and pupae stages (Table 4). The results can be explained by the fact that these EOs penetrate the chorion of the alternative host, *E. kuehniella*, causing lethal and/or sublethal effects on the parasitoid *T. pretiosum*. The greater tolerance of the egg-larvae stage of *T. pretiosum* compared to the pre-pupae and pupae stages may be due to the parasitoids mostly being in the egg stage. Thus, the chemical substances would have two barriers that are difficult to penetrate: the chorion of the alternative host egg and the

parasitoid. It must be considered that the chorion is composed of several layers and is an excellent barrier to prevent the entry of chemical substances since, during the evolution process, it was selected so that the embryo could breathe with limited water loss [72]. In the case of the pupa, it is a metabolically highly active phase in which a high rate of respiratory, protein, lipid, and carbohydrate metabolism can be observed [73] due to the histolysis and histogenesis that occur during this phase [74].

Regarding the selectivity of the EOs of *D. lanceolata* for the adult phase of *T. pretiosum*, the EO of the leaves was more selective regarding the longevity of adult females compared to the EO of the stem bark of this plant (Figure 8). However, for both EOs, it was found that females did not parasitize the eggs of the alternative host (Table 5). Based on data obtained in this work, it is impossible to affirm if the eggs of *E. kuehniella* treated with the EOs were rejected for oviposition of *T. pretiosum* or if these EOs have a repellent effect for adults of *T. pretiosum*. Therefore, these results open perspectives for further behavior studies with this species.

Although, in general, no physiological selectivity of the EOs from the leaves and stem bark of *D. lanceolata* was observed for the non-target organism *T. pretiosum*, under laboratory conditions, it is recommended to carry out further studies under semi-field and field conditions to study the selectivity of these EOs in situations of lower exposure of the natural enemy to substances, as recommended by the IOBC [34]. In addition, one can study the integration of these EOs with the adoption of techniques that allow ecological selectivity.

## 5. Conclusions

EOs from *D. lanceolata* (leaves and stem bark) showed insecticidal activity against FAW caterpillars in a low-dose topical application test. The insects showed rapid death after the application of the treatments. The major compounds in the EO of the leaves from *D. lanceolata* (β-caryophyllene and caryophyllene oxide) showed no toxicity when separately assayed. However, the mixture of β-caryophyllene and caryophyllene oxide reduced the probability of insect survival, but not in an identical way to the EOs. This result suggests that other substances present in the EOs from *D. lanceolata* (leaves), which were not identified in the present study, contributed to the insecticidal activity. Regarding selectivity, the EOs of the leaves from *D. lanceolata* were less toxic than the EOs of the stem bark of this plant against the immature phase of the parasitoid *T. pretiosum*. However, both EOs affected the parasitism of the females of *T. pretiosum*, requiring studies to evaluate ecological selectivity strategies for integrating these EOs.

**Supplementary Materials:** The following supporting information can be downloaded at: https://www.mdpi.com/article/10.3390/agriculture13020488/s1, Figure S1: Food preference of caterpillars (number of caterpillars) of *Spodoptera frugiperda* by essential oils from the leaves and stem barks of *Duguetia lanceolata* in the free-choice test. Ns = non-significant ($p \leq 0.05$) by $\chi^2$ test.

**Author Contributions:** Conceptualization, D.S.A., D.F.d.O., G.H.S. and G.A.C.; methodology, G.T.d.S.e.S., M.S.d.O., J.A.G.V., H.B.P., M.K.d.P.R., D.S.A., I.C.L., K.P. and A.S.S.; software, D.S.A., G.T.d.S.e.S. and M.S.d.O.; validation, G.T.d.S.e.S., M.S.d.O., J.A.G.V., H.B.P., M.K.d.P.R., D.S.A., I.C.L., K.P. and A.S.S.; formal analysis, G.T.d.S.e.S., M.S.d.O., J.A.G.V., H.B.P., M.K.d.P.R., D.S.A., I.C.L., K.P. and A.S.S.; investigation, G.T.d.S.e.S., M.S.d.O., J.A.G.V., H.B.P., M.K.d.P.R., D.S.A., I.C.L., K.P. and A.S.S.; resources, D.S.A., D.F.d.O., G.H.S. and G.A.C.; data curation, D.S.A. and G.H.S.; writing—original draft preparation, D.S.A., G.A.C., D.F.d.O., G.H.S. and G.T.d.S.e.S.; writing—review and editing, D.S.A., G.A.C., D.F.d.O., G.H.S. and G.T.d.S.e.S.; visualization, D.S.A., D.F.d.O., G.H.S. and G.A.C.; supervision, D.S.A., D.F.d.O., G.A.C. and G.H.S.; project administration, D.S.A., D.F.d.O., G.A.C. and G.H.S.; funding acquisition, D.S.A., D.F.d.O., G.A.C. and G.H.S. All authors have read and agreed to the published version of the manuscript.

**Funding:** This research was funded by Conselho Nacional de Desenvolvimento Científico e Tecnológico (CNPq) (Process CNPq 420598/2016-2; Process CNPq 429309/2018-07; Process CNPq 408121/2021-1), Coordenação de Aperfeiçoamento de Pessoal de Nível Superior (CAPES) (Scholarship), Fundação Araucária (FA) (APQ-02883-21), Superintendência Geral de Ciência, Tecnologia e

Ensino Superior (SETI) (Process PR 3793113741-3) and Fundação de Amparo à Pesquisa do Estado de Minas Gerais (FAPEMIG).

**Institutional Review Board Statement:** Not applicable.

**Data Availability Statement:** The data presented in this study are available on request from the corresponding author.

**Conflicts of Interest:** The authors declare no conflict of interest. The funders had no role in the design of the study; in the collection, analyses, or interpretation of data; in the writing of the manuscript; or in the decision to publish the results.

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
