# Peer review of "Duguetia lanceolata A. St.-Hil. (Annonaceae) Essential Oil: Toxicity against Spodoptera frugiperda (J. E. Smith) (Lepidoptera: Noctuidae) and Selectivity for the Parasitoid Trichogramma pretiosum Riley (Hymenoptera: Trichogrammatidae)"

_agriculture, doi:10.3390/agriculture13020488_

Round 1
Reviewer 1 Report
Rosetti et al. investigated effects of essential oils from Annona neolaurifolia, Duguetia lanceolata, and Xylopia brasiliensis, on the fall armyworm (FAW) Spodoptera frugiperda and its natural enemy, Trichogramma pretiosum. The authors investigated the chemical composition and content of essential oils extracted from different organs of three plants. They then examined the effects of extracts and some major compounds on both FAW and Trichogramma in several aspects. The paper is scientifically validated and well-written. I have only a few minor revisions to suggest.
The article examines the essential oils of three plants, but only one is mentioned in the title.
In some Figures, such as Fig 2, Fig 8, the font is too small to read.
In Table 4, it was not clearly indicated for each stage of Trichogramma.
Author Response
The authors greatly appreciate the contributions and time spent reviewing the manuscript. We believe that the suggestions will contribute to the improvement of the work. We try to make an effort to complete the modifications.
Reviewer 1 (question) - The article examines the essential oils of three plants, but only one is mentioned in the title.
Authors (response) – The authors have evaluated the toxicity of other plant species to Spodoptera frugiperda, however, we opted to include only the species Duguetia lanceolata in the title because it was the most promising and selected for the later stages of the study. Only the major compounds of this species were used in tests with the target insect S. frugiperda. Furthermore, only D. lanceolata was used in the selectivity assays with the egg parasitoid Trichogramma pretiosum. For this reason, we believe it is more appropriate to keep in the title only the name of the species D. lanceolata, which was the most explored in this study.
Reviewer 1 (question): In some Figures, such as Fig 2, Fig 8, the font is too small to read.
Authors (response) – The figures 2 and 8 have been redone with larger fonts.
Reviewer 1 (question): In Table 4, it was not clearly indicated for each stage of Trichogramma.
Authors (response) – Thanks for checking, it's been revised.
Best regards

Reviewer 2 Report
The work is interesting and presents an evaluation of biological activity of essential oils from selected plants for the management of Spodoptera frugiperda. Experiments are well described and scientifically sound.
Only the food preference experiment in my opinion required better description (and maybe using a graphical representation as well?).
Some of the tables reported in the figures are too small and hard to read. Please provide a better resolution.
EO analysis and % must be reported in triplicates showing the average abundance.
Overall this work is worth for publication on Agriculture.
More comments on the pdf attached.

Author Response
Dear reviewer,
The authors greatly appreciate the contributions and time spent reviewing the manuscript. We believe that the suggestions will contribute to the improvement of the work. We try to make an effort to complete the modifications.
Reviewer 2 (question): Only the food preference experiment in my opinion required better description (and maybe using a graphical representation as well?).
Authors (response): The authors are grateful for their contribution. We added a photo of the trial plot on line 205. We hope it has contributed to a better understanding.
Reviewer 2 (question): Some of the tables reported in the figures are too small and hard to read. Please provide a better resolution.
Authors (response): We redid some figures and believe we have improved the resolution.
Reviewer 2 (question): EO analysis and % must be reported in triplicates showing the average abundance.
Authors (response): The extracted materials were obtained from a single plant of each species. The harvested material was extracted at a single time, so that the oil represents a sample, representative of the plant. Further information that corroborates these statements can be found in the studies of:
CAN BAÅžER, K. Hüsnü; ÖZEK, Temel. Gas chromatographic analysis of essential oils. Gas Chromatography, p. 675–682, 1 jan. 2012.
MOHAMMADHOSSEINI, Majid et al. Profiling of the essential oil compositions from the flowers and leaves of Tanacetum fisherae Aitch. & Hemsl., an endemic plant in Kerman province, Iran. Natural Product Research, v. 36, n. 20, p. 5347–5352, 2022. Disponível em: <https://www-tandfonline.ez48.periodicos.capes.gov.br/doi/abs/10.1080/14786419.2021.1924711>.
KANT, R.; KUMAR, A. Review on essential oil extraction from aromatic and medicinal plants: Techniques, performance and economic analysis. Sustainable Chemistry and Pharmacy, Volume 30, December 2022, 100829.
Reviewer 2 (question): why don't test FAW eggs infested with the parasitoids?
Authors (response): FAW eggs are laid in the form of clusters; 3 and even 4 superimposed layers of eggs can be observed. The overlapping of egg layers prevents them from being previously unfeasible before testing with the parasitoid. If the eggs are not previously unfeasible, the experimental plot may be lost due to the hatching of the caterpillars. Thus, many researchers use the eggs of the alternative host as a model insect Ephestia kuehniella. E. kuehniella eggs are placed in isolation, facilitating handling and preventing them from being offered to parasitism beforehand.
Below you can find some articles in which the selectivity of products for Trichogramma spp. was studied using E. kuehniella as an alternative host.
COSTA, Mariana Abreu et al. Lethal, sublethal and transgenerational effects of insecticides labeled for cotton on immature Trichogramma pretiosum. Journal of Pest Science, v. 96, n. 1, p. 119–127, 1 jan. 2023. Disponível em: <https://link-springer-com.ez48.periodicos.capes.gov.br/article/10.1007/s10340-022-01481-9>.
LEITE, Germano L.D. et al. Nicosulfuron Plus Atrazine Herbicides and Trichogrammatidae (Hymenoptera) in No-Choice Test: Selectivity and Hormesis. Bulletin of Environmental Contamination and Toxicology, v. 99, n. 5, p. 589–594, 1 nov. 2017. Disponível em: <https://link-springer-com.ez48.periodicos.capes.gov.br/article/10.1007/s00128-017-2174-7>.
MOHAMMADHOSSEINI, Majid et al. Profiling of the essential oil compositions from the flowers and leaves of Tanacetum fisherae Aitch. & Hemsl., an endemic plant in Kerman province, Iran. Natural Product Research, v. 36, n. 20, p. 5347–5352, 2022. Disponível em: <https://www-tandfonline.ez48.periodicos.capes.gov.br/doi/abs/10.1080/14786419.2021.1924711>.
POTRICH, Michele et al. Is Isaria fumosorosea selective to Trichogramma pretiosum (Hymenoptera: Trichogrammatidae)? http://www.eje.cz/doi/10.14411/eje.2020.012.html, v. 117, n. 1, p. 110–117, 2020. Disponível em: <http://www.eje.cz/doi/10.14411/eje.2020.012.html>.
SOUZA, Jander R. et al. Toxicity of some insecticides used in maize crop on Trichogramma pretiosum (Hymenoptera, Trichogrammatidae) immature stages. Chilean Journal of Agricultural Research, v. 74, n. 2, p. 234–239, 2014.
Reviewer 2 (question): However, it must be considered that cypermethrin is a pure substance. - what do you mean? are you comparing cypermethrin (individual compound) to EOs (mixtrue of components)?
Authors (response): One of the importance of using a positive control is that it demonstrates that the population used is not resistant to a specific chemical group. Although it is more appropriate to use positive controls from the same chemical group, this is not always available. Regarding natural products, the only botanical pesticide registered in Brazil to control FAW is neem oil. However, it is known that this product does not have contact action for FAW. Thus, we chose to use cypermethrin as a positive control.
Reviewer 2 (question): what about using chemical standards? (line 126)
Authors (response): For a better understanding of the use of chemical standards, we have changed the text.
Reviewer 2 (question): was any particular behavior observed? (line 453)
Authors (response): As mentioned in line 461, the rapid death of insects can indicate an action in the nervous system.
Best regards

Reviewer 3 Report
This study investigated the toxicity of plant essential oils against the fall armyworm (FAW) and to explore the toxicity of plant essential oils to Trichogramma pretiosum, a natural enemy of the FAW. Although this work is extremely interesting as its findings contribute significantly to the development of botanical insecticides that can be combined with biological control agents to control the FAW, several questions remain in this paper:
1. Line 101 Plant samples collected need to be identified, please add the identification information.
2. Line 161-163 The references cited are not appropriate.
3. Line 315 Survival probability data need to add units.
4. Line 396,413 3.3.1. and 3.3.2. part of the title does not match the content.
Author Response
The authors greatly appreciate the contributions and time spent reviewing the manuscript. We believe that the suggestions will contribute to the improvement of the work. We try to make an effort to complete the modifications.
Reviewer 3 (question): Line 101 Plant samples collected need to be identified, please add the identification information.
Authors (response): The species were identified by a specialist (botanist) from Esal Herbarium, Federal University of Lavras. The information has been added to line 104.
Reviewer 3 (question): Line 161-163 The references cited are not appropriate
Thanks for checking. It's been revised.
Reviewer 3 (question): Line 315 Survival probability data need to add units.
Thanks for checking. The information has been added to line 317.
Reviewer 3 (question): Line 396,413 3.3.1. and 3.3.2. part of the title does not match the content.
Thanks for checking. The information has been corrected in lines 402 and 420.
Best regards
